# Image Enhancement Thanks to Negative Grey Levels in the Logarithmic Image Processing Framework

**DOI:** 10.3390/s24154969

**Published:** 2024-07-31

**Authors:** Michel Jourlin

**Affiliations:** Laboratoire Hubert Curien, UMR CNRS 5516, 18 Rue Professeur Benoît Lauras, 42000 Saint-Étienne, France; michel.jourlin@univ-st-etienne.fr

**Keywords:** image, enhancement, logarithmic image processing, negative grey level, full dynamic range

## Abstract

The present study deals with image enhancement, which is a very common problem in image processing. This issue has been addressed in multiple works with different methods, most with the sole purpose of improving the perceived quality. Our goal is to propose an approach with a strong physical justification that can model the human visual system. This is why the Logarithmic Image Processing (LIP) framework was chosen. Within this model, initially dedicated to images acquired in transmission, it is possible to introduce the novel concept of negative grey levels, interpreted as light intensifiers. Such an approach permits the extension of the dynamic range of a low-light image to the full grey scale in “real-time”, which means at camera speed. In addition, this method is easily generalizable to colour images and is reversible, i.e., bijective in the mathematical sense, and can be applied to images acquired in reflection thanks to the consistency of the LIP framework with human vision. Various application examples are presented, as well as prospects for extending this work.

## 1. Introduction

Because of its multiple applications, the problem of image enhancement has been widely studied and has given rise to numerous papers, of which it is impossible to present an exhaustive survey. One of the oldest and best-known techniques is based on histogram equalization: [1], and, more recently, [2]. Among the multitude of methods concerning spatial or frequency domains, let us mention the fuzzy approach [3,4], an algorithm based on Fourier Transform with Highpass (sharpening) and Lowpass (smoothing) Filters [5], a solution using gamma correction [6].

Recently, deep learning methods have addressed the issue of image enhancement for various applications such as autonomous driving [7], iris recognition [8], and underwater images [9,10]. In this last paper, a novel method was proposed based on an embedded fusion mechanism. The authors observed that the quality of the results depends on the quality of the input image, so they performed a pre-processing step to enhance the initial image quality based on the white balance algorithm and contrast-limited adaptive histogram equalization.

Despite the multitude of references, very recent papers show that the subject is still relevant. Among these, we have selected two papers that will be used to compare our approach to state-of-the-art methods. In Ref. [11], a novel Retinex-based Network (R2RNet) is presented, which includes specialized subnets dedicated to denoising and contrast enhancement. Compared to various enhancement techniques, R2RNet appears efficient for all degraded images. In Ref. [12], the authors present a “Cyclic Generative Attention-Adversarial Network” (CGAAN). This network is unsupervised and performs the enhancement of low-light images without using paired datasets. To improve the algorithm’s efficiency in terms of visual quality, the authors propose a novel adaptative attention mechanism under the feature maps, allowing the algorithm to focus on the most significant regions. The network is then guided to recover the high-quality images. The effectiveness of CGAAN will be compared to many methods that will be described and used in Section 6.3, including DALE (Dark region-Aware Low-light image Enhancement), DRBN (Deep Recursive Band Network), and DSLR (Deep Stacked Laplacian Restorer).

Finally, in Ref. [13], the interested reader will find a comprehensive review of low-light image enhancement in terms of network structure, training data, and the presentation of evaluation metrics.

Most of these methods are empirical and lead to irreversible transformations (mathematically non-bijective), resulting in a significant loss of information.

The common objective of enhancement methods is to best compensate for the effects of poor acquisition conditions (low-light images) or variable acquisition conditions (unstable illumination). Given the multitude of algorithms available, there are two pitfalls to overcome when evaluating the effectiveness of a given method and comparing it to others.

The first and most important one is to give, if possible, a non-empirical justification (mathematical, physical, or optical) for the considered approach. The present study focuses on this point of view.

The second is to implement objective tools to evaluate the “quality” of the result. Let us remark that it is illusory to try to create absolute criteria that are relevant in all situations. Let us give an example. In the case of breast cancer detection on mammography images, the observation of small white spots representing microcalcifications constitutes information that should be preserved at all costs but that could be considered noise in other situations. Authors generally refer to quality parameters, which are extremely numerous in the literature, and they are most often presented as quality metrics without always satisfying the mathematical properties required for a metric. The most used are the Peak Signal-to-Noise Ratio (PSNR) parameter and the Structural Similarity Index Measure (SSIM) parameter. The PSNR parameter is known to be poorly correlated with human visual appreciation, while the SSIM and its many variants take the human visual system into account a little better but are not highly significant.

For all these reasons, we chose to work in the LIP (Logarithmic Image Processing) framework, which is presented in the following section. The LIP framework allows us to define an enhancement algorithm, “Full Dynamic Range Expansion” (FDRE), based on the new concept of negative grey levels, which are interpreted as light intensifiers. The advantages of this method are multiple. FDRE enhances a low-light image without a loss of information because it is based on a mathematically bijective transformation. Moreover, this algorithm inherits the properties intrinsic to the LIP model: it has a strong physical justification based on the Transmittance Law, is consistent with the human visual system, and runs at the camera rate. Finally, it is also adaptable to images acquired in reflection and easily extended to colour images.

To conclude this introduction, let us note that the present study extends a previous publication we authored in the journal *Sensors* [14], in which low-light images were enhanced thanks to the ability of the LIP addition law to simulate variable exposure times. In the same paper, the problem of denoising enhanced images was discussed (CNN approach) and the quality of the results was evaluated in terms of the PSNR and SSIM; thus, this question will not be studied here.

## 2. Recalls of the LIP Framework

The LIP model was introduced in 1987 [15]. Readers interested in more information on the subject can refer to [16].

Let I(D, [0, M[) represent the set of grey level images defined on the same spatial support D with values in the grey scale [0, M[. In a first step, images belonging to I(D, [0, M[) are considered acquired in transmission, so that we can associate to a pair (f∈I(D, [0, M[), x∈D) the concept of transmittance $T_{f}\left(x\right)$, defined as the ratio of the out-coming flux at x by the incoming flux (intensity of the source). Mathematically, $T_{f}(x)$ represents the probability, for a particle of the source incident at x, to go through the obstacle, i.e., to be seen by the sensor.

**Remark** **1:***To simplify notations, we will confuse in a same letter “*f*” the grey level image and the semi-transparent object having generated* f.

Two laws were defined on the space I(D, [0, M[):-The addition law ⨹:

(1)$$f⨹g=f+g-\frac{f.g}{M}$$
which is a direct consequence of the Transmittance Law: $T_{f⨹g}=T_{f}\;.\;T_{g}$ that represents the probability for a particle of the source to pass through the superposition of two obstacles f and g.

-The scalar multiplication ⨻ is associated with a real number λ and an image f:



(2)
$$\lambda ⨻f=M-M{\left(1-\frac{f}{M}\right)}^{\lambda }$$



**Remark** **2:***In the context of images acquired in transmission, the grey scale* [0, M[ *is inverted. In fact, stacking obstacles between the source and the sensor obviously darkens the resulting image so that* M *represents the black extremity of the scale, which corresponds to a limited situation of opacity. Conversely, the white extremity* 0
 *is associated with a situation of transparency and represents the sensor’s response when it observes the source.*

Let us recall [15} that the two previous laws possess all the properties required to give I(D, [0, M[) a Vector Space structure, except the existence, for an image f, of an opposite g=⨺f.

Such an opposite must satisfy f⨹g=0, which seems a priori unfeasible: indeed, adding g to a given object f would make a transparent object!

However, according to Formula (1), the equation f⨹g=0 leads to the formal expression $⨺f=\frac{-f}{1-\frac{f}{M}}$, which takes its values in the interval ]−∞, 0].

Under these conditions, we define an over-space of I(D, [0, M[), noted $F\left(D,\right]-\infty ,\;M[)$, representing the set of functions defined on D with values in ]−∞, M[. We clearly have the inclusion $I(D,\;[0,\;M[)\subset F\left(D,\right]-\infty ,\;M[)$. It is important to note that an element f of $F\left(D,\right]-\infty ,\;M[)$ can be considered as a virtual image with possible negative grey levels.

## 3. Major Properties of LIP Laws

Most image-enhancement methods are not based on sound science. We chose the LIP framework because it complies with physical, optical, mathematical and psychovisual properties:-Strong mathematical structure

It is easy to verify that the space $F\left(D,\right]-\infty ,\;M\left[,\;⨹,\;⨻\right),$ satisfies all the conditions required (cf. [17], for example) to become a Real Vector Space. The consequences of such a remark are considerable: it gives access to countless concepts and properties introduced by mathematicians in the context of Vector Spaces, like interpolation, norms, and scalar products.

-Consistency with the Human Visual System

We first considered the case of images acquired in transmission. However, all the results obtained within the LIP framework apply to images acquired in reflection, thanks to a paper published by Brailean [18] in which he established the consistency of the LIP model with Human Vision. Under these conditions, LIP tools can be used to interpret images in the same way as the human eye would. 

-Simulation of variable exposure time

This property was presented by Carré et al. [19], “LIP operators: Simulating exposure variations to perform algorithms independent of lighting conditions”, and was improved in 2021 in *Sensors* [14], “Extending Camera’s Capabilities in Low Light Conditions Based on LIP Enhancement Coupled with CNN Denoising”.

-Simulation of variable thickness/opacity for images acquired in transmission

Such a property is useful in a variety of situations.

One example is the case of physical cuts performed with a microtome, where regularity in the thickness of successive slices is difficult to achieve. When acquiring images of such slices, variations in thickness lead to variations in opacity, which must be corrected to obtain homogeneous 3D reconstructions.

Another case is confocal microscopy. When acquiring images of a semi-transparent object (human skin and hair) at different depths of focus, the brightness of the images decreases sharply as the depth increases, requiring a step to compensate for this attenuation in brightness.

-Changing the dynamic range of an image

This topic is developed in the following sections.

## 4. Image Enhancement with LIP Laws: A Quick Reminder

For images acquired in transmission, it is easy to explain the darkening/brightening effect of each law (cf. Figure 1 and Figure 2).

Law ⨹: If C∈[0, M[ and f∈I(D, [0, M[, calculating f⨺C (resp. f⨹C) consists of subtracting (resp. adding) a uniform image C from (resp. to) f. Considering a very low-light image f, to obtain a satisfactory enhancement of f, the constant to be subtracted from f takes values close to M, at the risk of observing negative values of f⨺C. Such a remark draws our attention to a physical interpretation of negative grey levels (cf. next section). 

Law ⨻: If λ is a real number and f∈I(D,[0,M[, calculating λ⨻f consists of multiplying the thickness of f by λ, which obviously produces a brightening effect when λ∈]0,1[ and a darkening effect when λ∈]1,+∞[, but not always an increase in the dynamic range of f.

Looking at Figure 1 and Figure 2, it is easy to imagine the potential effectiveness of LIP laws for image enhancement, which explains the numerous publications on the subject. For the record, we will just mention the initial work of two teams led by Professor Cahill (La Trobe University, Australia) and Professors Agaian and Panetta (respectively, the University of Texas and Tufts University, USA). In [20], Deng and Cahill describe a new implementation of Lee’s image enhancement algorithm. Based on the Logarithmic Image Processing (LIP) model, the proposed approach can simultaneously improve the overall contrast and sharpness of an image. In [21], Agaian and Panetta introduce a parameterized LIP (PLIP) model that covers both the linear arithmetic and LIP operations within a single unified model.

Image enhancement is usually performed by optimizing some parameter, such as the dynamic range or contrast concepts. It should be noted that the dynamic range must be treated with caution. Indeed, the presence of a single white pixel and a single black pixel in an image produces a maximum dynamic range without guaranteeing an interesting visual result.

For this reason, another approach was proposed in [16] for a low-light image f: it consists of calculating either the constant $C_{0}$, maximizing the histogram standard deviation $\sigma [h\left(f⨺C\right)$ of f⨺C (cf. Figure 3), or $the\;real\;number\;\lambda_{0},\;maximizing\;\sigma [h\left(\lambda ⨻f\right)$ (cf. Figure 4), with the constraint, in both cases, of staying within the grey scale.

## 5. Optical Interpretation of Negative Grey Levels

In this section, we will take advantage of the Vector Space Structure of the space of functions $F\left(D,\;\right]-\infty ,\;M[$) thanks to logarithmic laws ⨹ and ⨻. Moreover, this structure allows a well-founded optical interpretation of negative grey levels. Indeed, when we introduced the opposite $⨺f\left(x\right)=\frac{-f(x)}{1-\frac{f(x)}{M}}$ of a grey level $f\left(x\right),$ our goal was that equality $\left(⨺f(x)\right)⨹f(x)=0$ be satisfied. This means that the union of the two “obstacles” ⨺f(x) and $f\left(x\right)$ becomes transparent, as if the sensor was observing the source!

Thus, ⨺f(x) can be interpreted as a light generator that increases the intensity of emitted “photons” (or electrons, or X-rays depending on the source elements) to produce an outgoing flux at x equal to that of the initial source. In Figure 5, we present the grey level values along a straight line drawn on the considered image of a chart and their opposites in the interval ]−∞, 0[ consisting of negative grey levels.

Similarly, the scalar multiplication λ⨻f can be considered for negative values of λ: the previous interpretation in terms of thickness variation adapts to negative values thanks to the concept of “negative” thickness, which acts as a brightening operator.

To formalize this approach in optical terms, let us define the Generalized Grey Scale ]−∞, M[ representing the set of all possible values of source intensity: the value −∞ corresponds to a theoretical infinite intensity, while M corresponds to a null one.

Choosing an initial source intensity $I_{S}$ ∈]−∞, M[ involves selecting an origin $0_{S}$ for the grey scale ${[0}_{S},\;M[$, producing the current space of images $I(D,\;[0_{S},M[)$. The over-space $F\left(D,\;\right]-\infty ,M[)$ then represents the set of all possible images when the reference source moves inside the interval ]−∞,M[, which leads to the following formula:(3)$$F\left(D,\;\right]-\infty ,M[)=\;{⋃\;}_{0_{S\;}\in ]-\infty ,M[}I(D,\;[0_{S},M[)$$

The following section presents the ability of negative grey levels to perform a maximum dynamization of a very-low-light image.

## 6. Full Dynamic Range Expansion (FDRE) Algorithm

For an image f, its dynamic range DR (f) is calculated as DR(f)=f(a)−f(b), where
(4)$$f\left(a\right)=Sup\;\left\{f\left(x\right),\;x\in D\right\}\;and\;f\left(b\right)=Inf\;\left\{f\left(x\right),\;x\in D\right\}$$

Case: subtraction of a constant

From the formula, $f\left(x\right)⨺C=\frac{f\left(x\right)-C}{1-\frac{C}{M}}$, the dynamic range DR (f⨺C) obviously satisfies $DR\;(f⨺C)=\frac{f(a)-f(b)}{1-\frac{C}{M\;}}$ and increases with C when C∈[0, M[. Moreover, $DR\;\left(f⨺C\right)$ tends toward +∞ when *C* approaches *M* because $1\;–\frac{C}{M}$ tends toward *0*. Thus, a value

$C_{0}$ can be calculated maximizing $DR\;\left(f⨺C\right)$, i.e., satisfying

$DR\;(f⨺C_{0})=M-1$, which corresponds to the maximum range for digitized images.

**Remark** **3:***Considering that*$f\left(x\right)⨺C$*is expressed as an affine function*α.f(x)+β*, the proposed full expansion corresponds to a classical linear expansion, which consists of applying a bijection of the observed dynamic range* [f(b), f(a)] *onto the whole grey scale* [0, M−1]. 

Case: scalar multiplication

When negative values of λ are allowed, the function λ⨻f takes its values inside the interval ]−∞, 0], and when λ varies from 0 to −∞, DR(λ⨻f) varies from 0 to +∞. Under these conditions,

DR(f)=f(a)−f(b) and for λ < 0:(5)$$DR\left(\lambda ⨻f\right)=\lambda ⨻f\left(b\right)-\lambda ⨻f\left(a\right)=M-M{\left(1-\frac{f\left(b\right)}{M}\right)}^{\lambda }-\left[M-M{\left(1-\frac{f\left(a\right)}{M}\right)}^{\lambda }\right]\;DR\left(\lambda ⨻f\right)=M\left[{\left(1-\frac{f\left(a\right)}{M}\right)}^{\lambda }-{\left(1-\frac{f\left(b\right)}{M}\right)}^{\lambda }\right]$$

It is important to note that there exists a unique value $\lambda_{0}<0$ such that $DR\left(\lambda_{0}⨻f\right)=M-1.$ Then,
(6)$$M\left[{\left(1-\frac{f\left(a\right)}{M}\right)}^{\lambda_{0}}-{\left(1-\frac{f\left(b\right)}{M}\right)}^{\lambda_{0}}\right]=M-1\;\Leftrightarrow \;{\left(1-\frac{f\left(a\right)}{M}\right)}^{\lambda_{0}}-{\left(1-\frac{f\left(b\right)}{M}\right)}^{\lambda_{0\;}}=1-\frac{1}{M}$$

To achieve the *FDRE* of f, we must perform the two following steps:

Step 1: Computation of $\lambda_{0}$

In Equation (6), it is not possible to calculate $\lambda_{0}$ explicitly. Nevertheless, we can assert the uniqueness of $\lambda_{0}$. Indeed, the dynamic range $DR(\lambda_{\;}⨻f)$ is expressed as the difference between two exponential functions and is then continuous with respect to the variable λ, meaning that it reaches each value lying in [0,+∞[ only once—in particular, the value $1-\frac{1}{M}$ when $\lambda_{\;}=\lambda_{0\;}.$

To obtain an approximate solution, we perform a dichotomy step to reduce an interval surrounding $\lambda_{0\;}$. First, we choose the initial size of the interval, say 1, and determine the first value λ=−1,−2,−3… for which the following inequality applies:(7)$${\left(1-\frac{f\left(a\right)}{M}\right)}^{\lambda }-{\left(1-\frac{f\left(b\right)}{M}\right)}^{\lambda }>1-\frac{1}{M}$$

This means that the optimal value $\lambda_{0}$ belongs to the interval ]λ, λ+1]. The principle of dichotomy consists then of dividing this interval by 2 depending on the location of $\lambda_{0}$ in $]\lambda ,\;\lambda +\frac{1}{2}]$ or $]\lambda +\frac{1}{2},\;\lambda +1]$. The operation is iterated until the required precision for $\lambda_{0}$ is reached.

Step 2: Displaying the FDRE $\lambda_{0\;}⨻f-(\lambda_{0\;}⨻(f\left(a\right))$ of image f

Once $\lambda_{0}$ computed, the function $\lambda_{0\;}⨻f$ must be displayed as an image. This is achieved by translating $\lambda_{0\;}⨻f$ by the vector $-(\lambda_{0\;}⨻(f\left(a\right))$, which results in a Full Dynamic Range Expansion of f. In Figure 6, two examples are presented:

The first one deals with the centrifugation of blood to separate red blood cells (erythrocytes) from plasma. The aim is to evaluate the position of the boundary between the two media on an image acquired in back-light condition, which is fairly easy, but also to be able to “read” the information on the label stuck to the test tube, in particular the batch reference and barcode. The second concerns the “Indoor scene” image in Figure 4.


**Comments:**



*The FDRE technique is easily performed in real time (31 frames per second) for images of standard a size of 512 × 512 pixels.*



*The FDRE technique can be extended to colour images in a very simple way.*


### 6.1. FDRE for Colour Images

We associate to a colour image f its components $f_{R},\;f_{G},\;f_{B}$ in the three channels Red, Green, and Blue, and we define the Global Dynamic Range $GDR\left(f\right)$ of image f according to
(8)$$GDR\left(f\right)={Max}_{x\in D}\left\{f_{R}(x),\;f_{G}(x),\;f_{B}(x)\right\}-{Min}_{x\in D}\left\{f_{R}(x),\;f_{G}(x),\;f_{B}(x)\right\}$$

To apply the FDRE algorithm to a low-light colour image f, we compute the unique negative scalar $\lambda_{0}$ satisfying
(9)$$GDR\left(\lambda_{0\;}⨻f\right)=M-1$$

Multiplying each of the three components $f_{R},\;f_{G},\;f_{B}$ by $\lambda_{0}$ results in three negative functions $\lambda_{0}⨻f_{R},\;\lambda_{0}⨻f_{G},\;\lambda_{0}⨻f_{B}$. To bring these functions back into the grey scale [0,M[, we apply them a translation by the vector $-\lambda_{0}⨻\left[{Max}_{x\in D}\left\{f_{R}\left(x\right),\;f_{G}\left(x\right),\;f_{B}\left(x\right)\right\}\right]$. Examples are given in Figure 7.

### 6.2. Physical Interpretation of the Proposed FDRE Algorithm

As recalled in Section 3, the LIP framework is based on the Transmittance Law; the acquisition of an image inside this model is performed according to Figure 8.

The logarithmic addition (resp. subtraction) of a constant C to (resp. from) an image f can be represented in the same way (Figure 9).

Moreover, the LIP framework is consistent with the Human Visual System; its applicability is not limited to images acquired in transmission but naturally extends to images acquired in reflection that we wish to analyse as a human eye would.

To provide a well-founded optical interpretation of the FDRE algorithm, including the case of an image f acquired in reflection, the grey level f(x) of each pixel x is considered. In the LIP grey scale, where 0 represents the maximum intensity and M is the total opacity, f(x) obviously appears as an intermediate value between “white” and “black”, which means as an attenuation of the source, interpreted as a transmittance. It is then possible to design a virtual semi-transparent obstacle producing for each pixel x the value f(x). Under such conditions, the computation of $\lambda_{0}$ corresponds to a theoretical negative thickness of this semi-transparent object to reach the maximum available dynamics, that is, the dynamics of the full grey scale.

### 6.3. FDRE Algorithm’s Efficiency Compared with Other Methods

Any author of a novel image enhancement algorithm has the legitimate ambition to evaluate its performance and compare it to existing methods in terms of visual quality. For this purpose, we could refer to classical parameters (PSNR, SSIM), but we know that such parameters are not strongly consistent with the human visual system. Moreover, it is illusory to hope to create a “universal” parameter suitable for all situations, like very-low-light images or dark images with bright regions, all being potentially acquired under varying light conditions. This is why the authors interested in image enhancement are generally satisfied with subjective assessments. In the following Section 6.3.1 and Section 6.3.2, we compare FDRE with various algorithms described in [11,12] cited in the introduction.

#### 6.3.1. Comparison with SRIE, RetinexNet, and R2RNet

In Section 1, we presented the R2RNeT method proposed by Hai et al. [11]. In this study, R2RNeT is compared (see Figure 10) to two classical enhancement algorithms: Simultaneous Reflectance and Illumination Estimation (SRIE) (cf. [22]) and Retinex-Net (cf. [23]):

SRIE: To estimate reflectance and illumination from a given image, the authors proposed a novel weighted variational model based on Simultaneous Reflectance and Illumination Estimation (SRIE). Compared to conventional variational models, SRIE together reduces noise and preserves the estimated reflectance with more details.

Retinex-Net: The authors collected a low-light dataset containing low/normal-light image pairs and developed a deep Retinex-Net learned on this dataset. This Retinex-Net includes a Decom-Net for decomposition and an Enhance-Net for illumination correction.

**Remark:** 
*Retinex theory, namely, the theory of the retinal cortex, established by Land and McCann, is based on the perception of colour by the human eye and the modeling of colour invariance [24].*


In Figure 11, we display the result of our FDRE method.

**Comment:** *Let us recall that the FDRE algorithm corresponds to a logarithmic expansion within the negative part of the grey scale (see Figure 5). Such an expansion acts less strongly on the darkest pixels than on the brightest ones, producing a visual appearance near saturation for the brightest regions (“Indoor scene” in Figure 7). It is important to note that no post-processing was applied to the image in Figure 11, unlike the image presented in Figure 10d after a denoising step.*

#### 6.3.2. Comparison with DALE, DRBN and CGAAN

In this section, we return to the algorithm “Cyclic Generative Attention-Adversarial Network” (CGAAN) presented in the introduction. Proposed by Zhen et al. [12], CGAAN is compared by the authors to two enhancement algorithms:

The novel enhancement method presented in [25] is called Dark region-Aware Low-light image Enhancement (DALE). It consists of applying a visual attention module to detect dark regions and apply a brightness enhancement. Such an approach preserves the colour of original images and normally avoids the saturation of illuminated regions.

A Deep Recursive Band Network (DRBN) is proposed in [26]. Based on paired low/normal-light images, it aims at recovering a linear band representation of an enhanced normal-light image. The band recomposition is learned with the perceptual guidance toward fitting perceptual regularization of high-quality images.

In Figure 12, we display the results of DALE, DRBN, and CGAAN (Figure 12b–d) applied to an initial low-light image (Figure 12a), and the result of our FDRE algorithm is proposed in Figure 12e.

To conclude this section, let us propose a comparison between a classical histogram equalization (HE) algorithm and the FDRE algorithm. In Figure 13, two low-light colour images are considered: “Indoor scene” and “Beach by night”. Classically, the HE algorithm is performed on the V-channel of their HSV representations.

We observe that on the very dark image, “Indoor scene”, HE produces false colours, while FDRE tends to saturate bright areas, as expected. Concerning “Beach by night”, which presents dark and bright regions, HE is ineffective, while FDRE highlights the information in the darkest areas.

## 7. Examples of Application

### 7.1. 3D Visualization of Scanner Dental Optical Cuts

The initial data consist of twelve slices of a jaw obtained using an X-ray scanner. To avoid over-irradiating the patient, dentists applied a low dose of X-rays. The resulting images therefore possess a limited contrast, requiring a FDRE step, based on scalar multiplication law ⨻, producing enhanced slices.

A major problem is raised by the scanner resolution anisotropy: the spatial resolution inside a slice is around 0.2 mm per pixel, while the thickness of each slice is around 1 mm. To overcome such a drawback, we applied a LIP interpolation computing four intermediate images between two successive slices to obtain cubic voxels. The formula for performing such an interpolation is very simple and deducted from a classical interpolation, written with LIP operators. We define the segment [f,g] associated with a pair of successive slices f and g according to
(10)$$\left[f,g\right]=\;{\left\{\left(\lambda ⨻f\right)⨹((1-\lambda )⨻g)\right\}}_{\lambda \in [0,1]}$$

In our situation, we used $\lambda =\frac{1}{5},\;\frac{2}{5},\;\frac{3}{5},\;\frac{4}{5}$ (cf Figure 14) to compute the four intermediate slices.

Since the initial slices are acquired in transmission, the LIP interpolation is perfectly adapted and produces intermediate grey levels in the sense of transparency. The grey level of a pixel belonging to slice f is assigned to the corresponding voxel.

Finally, we dispose of 56 images instead of the first 12.

In Figure 15a, the initial twelve under-lighted slices are displayed, as well as their anisotropic superposition. Applying the FDRE algorithm produces enhanced images (Figure 15b), highlighting the location of an included tooth considered the target.

The expectations of dentists were multiple and rather difficult to meet.

Their ultimate objective was to visualize in 3D all the optical sections of the scanner to improve their understanding and prepare for surgical procedures. For that, it is necessary to perform an automated segmentation of this 3D block whose segments correspond to the anatomical elements such as teeth, mandibular bone, and sinuses. In Figure 15c, a fairly simple example is proposed. In fact, it is not difficult to remove from the 3D block the regions corresponding to air and flesh, which are the darkest ones. This step is performed using a classic Region Growing algorithm.

Once an angle of view of the 3D block is chosen, the voxels located in the foreground are visualized with their grey level, while the others are not considered.

It is then easy to estimate the precise location of an included tooth not initially visible because it is entirely under the gum surface and thus to perform the minimum incision to achieve the extraction.

### 7.2. Other Fields of Application

Logarithmic image enhancement is applicable in many areas. We can cite the following:

- Thickness normalization of serialized slices made using a microtome or an ultra-microtome to standardize the dynamics of the corresponding images. This allows us to visualize homogeneous 3D blocks.

- Brightness correction of images acquired using a confocal microscope to observe semi-transparent objects (for example, skin and hair). Obviously, the dynamics of an image corresponding to a deep focal plane are strongly attenuated. In Figure 16, the acquisition of a human hair using a Tandem Scanning Confocal Microscope (TSM) is presented. We know that such a microscope can acquire the relief of an object. In fact, a pixel located at the object surface is assigned a maximum grey level when this pixel belongs to the focal plane of the microscope.

In Figure 16a, each pixel of the hair relief map is displayed with its maximum reflectance. When the focal plane of the TSM crosses the hair near its axis, we obtain a very low-light image of the medulla. After FDRE enhancement (Figure 16b), the segmentation of the medulla is performed thanks to a Region Growing algorithm that detects the hair boundaries and their grey levels together. Finally, the registration of such images, when the focal plane of the TSM moves through the entire thickness of the hair, allows a 3D visualization of the medulla (in false colour, Figure 16c) together with the hair surface seen from the inside.

## 8. Conclusions

This study focuses on low-light image enhancement, which has long been studied by many researchers with very different approaches that are often empirical. Our stated goals were to propose novel algorithms with a strong physical justification, consistent with the human visual system, with the shortest possible execution time. It has been demonstrated that the Logarithmic Image Processing (LIP) framework meets all these constraints and permits the introduction of negative grey levels to simulate an increase in the source intensity. Knowing that the logarithmic laws ⨹ and ⨻,respectively, model a variation of exposure time and a variation of opacity (thickness) of the observed object, we dispose of scientifically based and efficient tools to extend the dynamics of low-light images to the full dynamic range (FDRE algorithm). Due to the non-linearity of the logarithmic laws, FDRE acts less strongly on the darkest pixels than on the brightest ones, producing a visual aspect near saturation for the brightest regions. In conclusion, FDRE will achieve maximum efficiency for very-low-light images or regions.

In addition, the consistency of the model with the human visual system augurs for satisfactory results in terms of visual quality, while such an evaluation is not guaranteed by the usual “metrics” PSNR and SSIM. Moreover, the novel algorithm runs at a conventional camera speed (24 frames per second) using a standard PC.

## 9. Perspectives

Our goal now is to strengthen the relevance of the proposed tools by creating specialized quality evaluators, such as sharpness, contrast and resolution. The first criterion (sharpness) is particularly interesting, as it concerns different situations such as defocusing or the presence of a diffusing medium, as well as the application of Super-Resolution (SR) algorithms. By creating intermediate pixels, like any interpolation technique, SR algorithms require a de-blurring step, the effectiveness of which needs to be assessed. Since the blurring effect occurs near the contours (transition pixels), various papers have proposed quality criteria based on gradient operators. The problem is that most of these operators do not produce values naturally limited to the available grey scale but require adjustments that distort the information. We plan to use the notion of logarithmic contrast (LIP difference in terms of grey levels) between a pixel and its neighbors, with the advantage of remaining inside the grey scale. The initial results are promising and allow us to reclassify in the right order different defocused images of the same object. Moreover, such contrast concepts seem relevant to assess whether an image is well contrasted in the sense of human vision. It will be interesting to compare this approach with Artificial Intelligence methods concerning, for example, the learned similarity (cf. [27]).

## Figures and Tables

**Figure 1 sensors-24-04969-f001:**
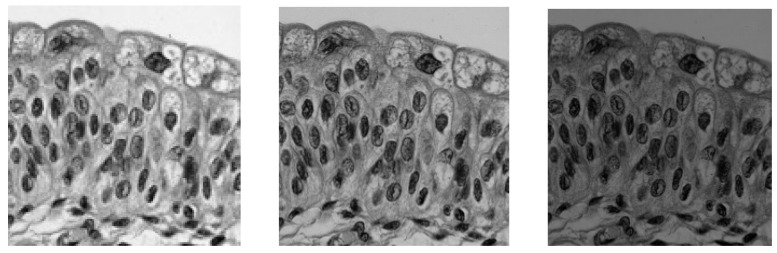
LIP-Subtraction (resp. Addition) of a constant from (resp. to) an image f, “Epithelial cells”. From left to right: visualization of f⨺C*, f,* and f⨹C, case *C* = *64* grey levels.

**Figure 2 sensors-24-04969-f002:**
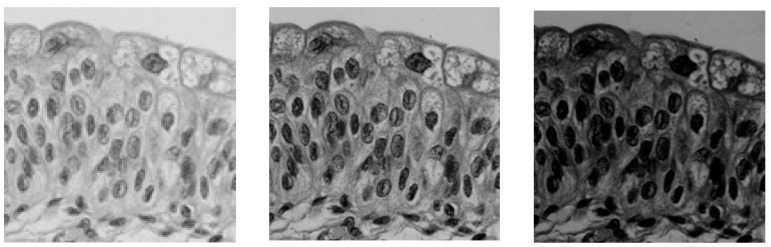
LIP-Multiplication by a real number. Image f,“Epithelial cells”. From left to right: visualization of *0.5*⨻f*,*
f, and *2*⨻f.

**Figure 3 sensors-24-04969-f003:**
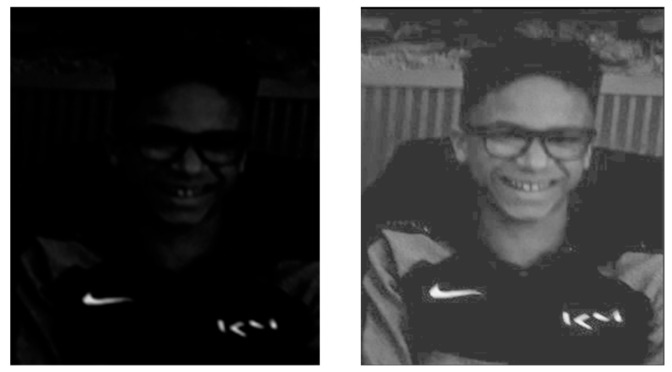
Maximization of the standard deviation $\sigma [h\left(f⨺C\right)$. From left to right: initial low-light image *f* “Teenager” (exposure time of 8 ms). Enhanced image of f according to $f⨺\;C_{0}$*, with*
${C_{0}}_{\;}=197$.

**Figure 4 sensors-24-04969-f004:**
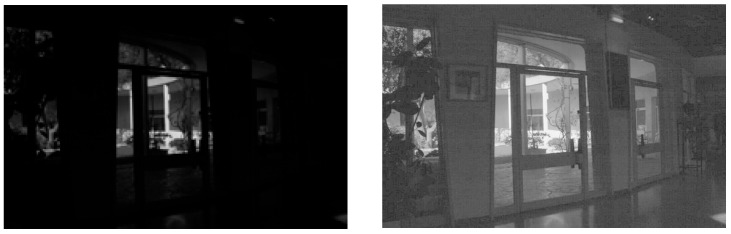
Maximization of the standard deviation $\sigma [h\left(\lambda ⨻f\right)\;$ of the histogram of λ⨻f. From left to right: Initial low light image *f,* “Indoor scene”, in the presence of bright points. Enhanced image of *f* according to $\lambda_{0}⨻f$, with ${\lambda_{0}}_{\;}=0.28$.

**Figure 5 sensors-24-04969-f005:**
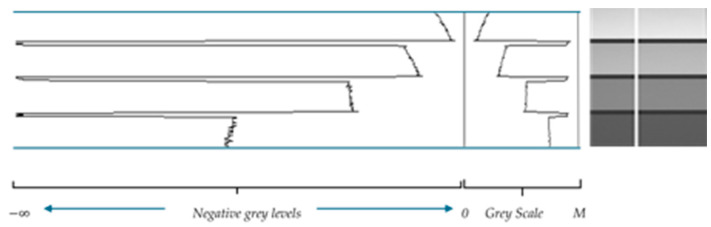
Grey levels along the white straight line drawn on the chart and their opposites.

**Figure 6 sensors-24-04969-f006:**
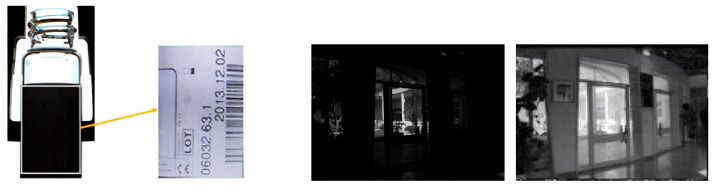
FDRE examples. From left to right: The test tube after centrifugation with the label location and the result of the LIP-multiplicative FDRE. Initial image “TIndoor scener” before and after multiplicative FDRE.

**Figure 7 sensors-24-04969-f007:**
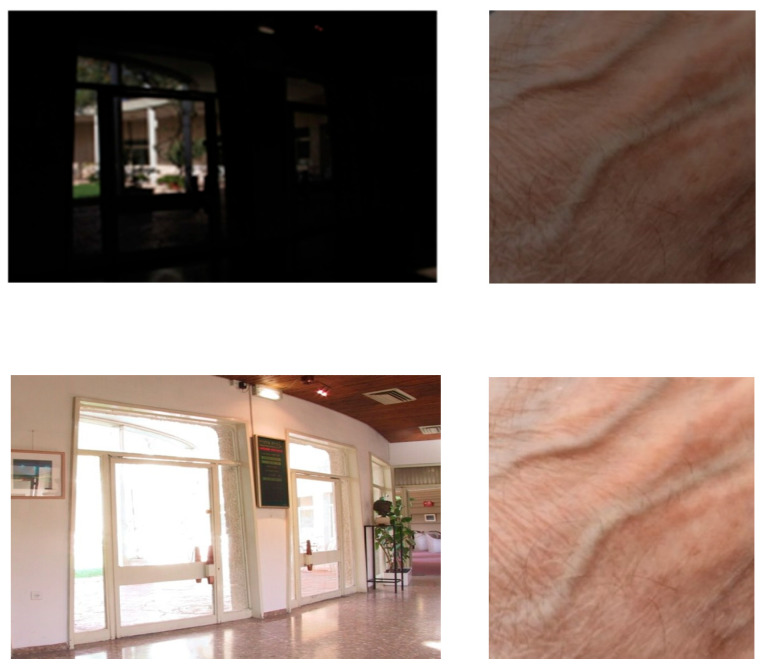
Low-light colour images f, g and their enhanced images $\lambda_{0}⨻f$, $\lambda_{0}⨻g$. From left to right: indoor scene, human skin.

**Figure 8 sensors-24-04969-f008:**
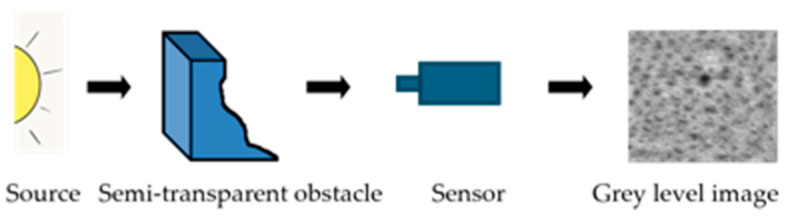
Image acquisition in transmission.

**Figure 9 sensors-24-04969-f009:**
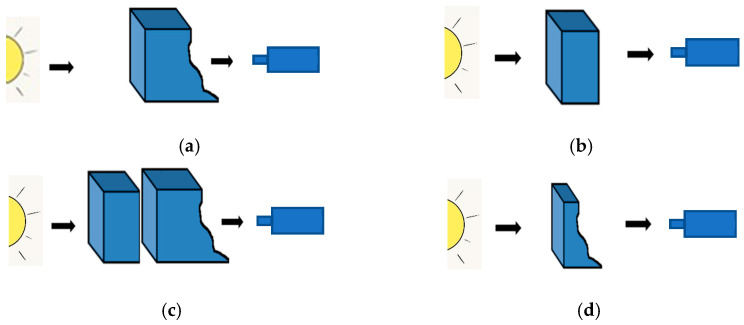
Representation of f, *C*, f⨹C, and f⨺C in transmission. (**a**) Acquisition of initial image *f*. (**b**) Uniform image *C*. (**c**) Acquisition of f⨹C. (**d**) Acquisition of f⨺C.

**Figure 10 sensors-24-04969-f010:**
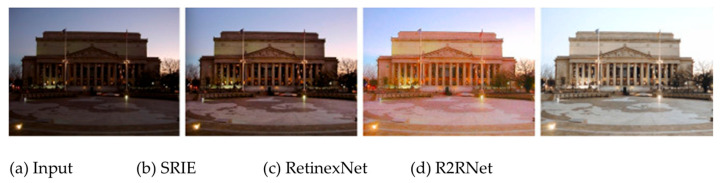
From a previous paper by Jiang Hai et al. [11]: (**a**) Input low-light image. (**b**) Simultaneous Reflectance and Illumination Estimation (SRIE) result (cf. [22]). (**c**) RetinexNet result (cf. [23]). (**d**) Result of the proposed algorithm.

**Figure 11 sensors-24-04969-f011:**
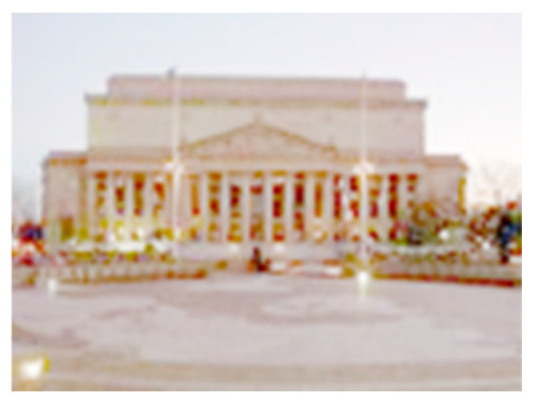
Result of FDRE, without any denoising post-processing.

**Figure 12 sensors-24-04969-f012:**
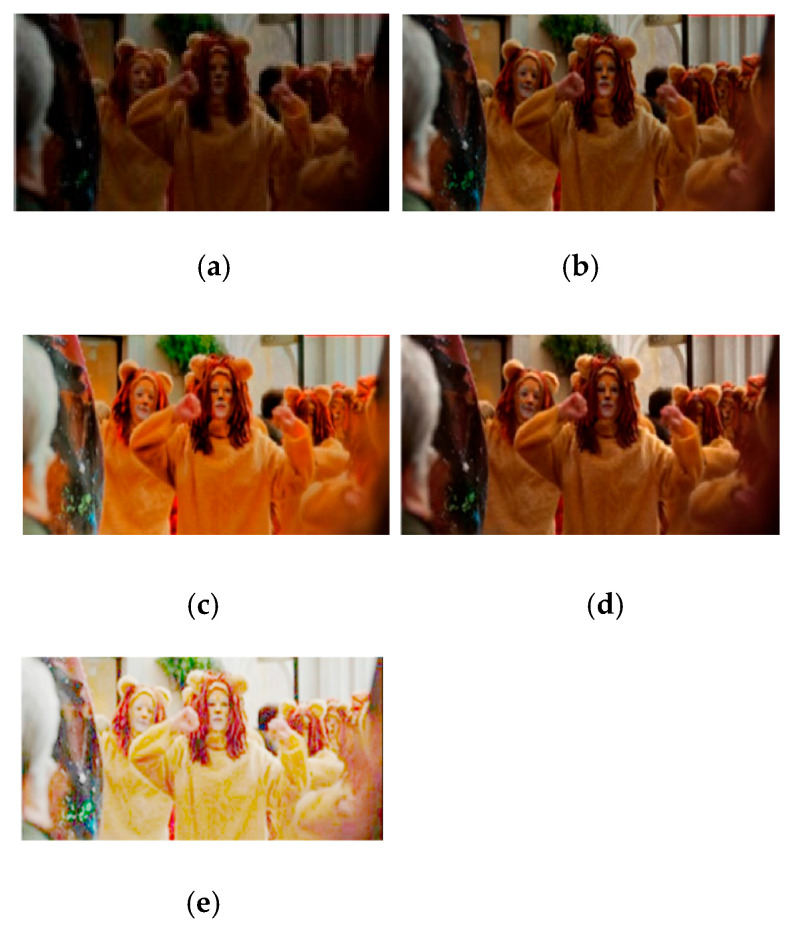
(**a**) Input, (**b**) DALE, (**c**) DRBN, (**d**) CGAAN, (**e**) FDRE.

**Figure 13 sensors-24-04969-f013:**
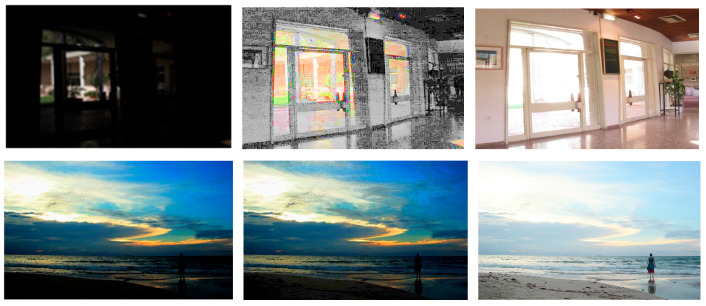
Top: from left to right, initial image “Indoor scene”, histogram equalization, FDRE. Bottom: from left to right, initial image “Beach by night”, histogram equalization, FDRE.

**Figure 14 sensors-24-04969-f014:**
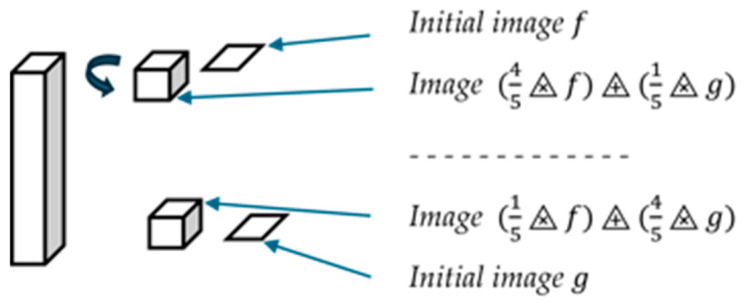
From left to right: initial voxel, interpolated voxels.

**Figure 15 sensors-24-04969-f015:**
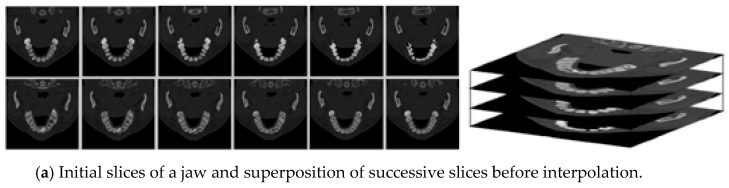
Based on 12 X-rays slices of a jaw, the different steps to perform a 3D visualization.

**Figure 16 sensors-24-04969-f016:**
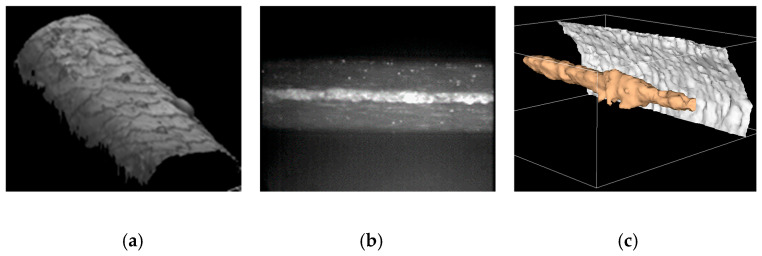
Confocal microscopy: visualization of a human hair. (**a**) Surface visualization. (**b**) Hair medulla after FDRE. (**c**) Medulla 3D visualization.

## Data Availability

Ethical review and approval were waived for this study. In fact, the only concerned section (Section 7.1) entitled “3D Visualization of Scanner Dental Optical Cuts” concerns private data (of family origin).

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
