# Peer review of "Image Enhancement Thanks to Negative Grey Levels in the Logarithmic Image Processing Framework"

_sensors, 2024, doi:10.3390/s24154969_

Round 1

Reviewer 1 Report

Comments and Suggestions for Authors

The manuscript dealt with the LIP framework by providing a physical justification for the introduction of negative grey levels, in particular to simulate an increase of the source intensity. Knowing that the  logarithmic laws model a variation of exposure time and a variation of opacity (thickness) of the observed object. The authors dispose of scientifically based and efficient tools to extend the dynamics of low-light images. In addition, the consistency of the model with the human visual system augurs for satisfactory results in terms of visual quality, while such an evaluation is not guaranteed by the usual "metrics" PSNR and SSIM.

However, there are minor correction I have observed.

Title:

No abbreviations expect it is well know one: Please write in full "Logarithmic Image Processing" = (LIP) 

Abstract:

The abstract is not well written. Authors rearrange it to follow this in a concise manner: 

Introduction

Problem Statement

Aim

Methodology

Results (be specific and let it be measurable)

Conclusion 

1. Introduction

Line 30 remove the dots ........

Reference:

The references are not recent. Review and include recent work (within the last five years) in your citations and reference.  

Comments on the Quality of English Language

The quality of English in this paper is of good standard but required minor editing 

Author Response

COMMENTS and SUGGESTIONS:

The manuscript dealt with the LIP framework by providing a physical justification for the introduction of negative grey levels, in particular to simulate an increase of the source intensity. Knowing that the  logarithmic laws model a variation of exposure time and a variation of opacity (thickness) of the observed object. The authors dispose of scientifically based and efficient tools to extend the dynamics of low-light images. In addition, the consistency of the model with the human visual system augurs for satisfactory results in terms of visual quality, while such an evaluation is not guaranteed by the usual "metrics" PSNR and SSIM.

However, there are minor correction I have observed.

Title:

No abbreviations expect it is well know one: Please write in full "Logarithmic Image Processing" = (LIP) 

Abstract:

The abstract is not well written. Authors rearrange it to follow this in a concise manner: 

Introduction

Problem Statement

Aim

Methodology

Results (be specific and let it be measurable)

Conclusion 

  1. Introduction

Line 30 remove the dots ........

Reference:

The references are not recent. Review and include recent work (within the last five years) in your citations and reference.  

Comments on the Quality of English Language

The quality of English in this paper is of good standard but required minor editing 

Submission Date

24 April 2024

Date of this review

20 May 2024 16:53:56

ANSWERS:

Many thanks for your relevant comments and remarks.

The suggested corrections have been done.

The abstract has been rewritten.

Recent references have been added (ref [10], [11], [12]).

Moreover, a novel section (6.3) has been dedicated to coimparison with existing methods.

Reviewer 2 Report

Comments and Suggestions for Authors

In this paper, the authors emphasize the compatibility of LIP with human vision for enhancing imagery.

My comments:

Figure 5 needs a more detailed explanation.

The authors should compare the LIP results in terms of imagery enhancements and processing time with the use of filters or arithmetic operations on channel values.

The 3D visualization of scanner dental optical cuts is a valuable application of LIP. However, the procedure for creating this visualization requires a clearer explanation.

The article outlines two main objectives, which should be explicitly stated as achieved in the conclusion.

The authors must divide the conclusion and perspectives section into discussion and conclusion sections.

Author Response

COMMENTS AND  ANSWERS

Comment 1; In this paper, the authors emphasize the compatibility of LIP with human vision for enhancing imagery.

Answer 1: First of all, many thanks for your relevant comments and remarks which helped me to improve the relevance and readability of the manuscript.

Comment 2: Figure 5 needs a more detailed explanation.

Answer 2: Figure 5 was incomplete in your manuscript. It has been modified and simplified for better understanding.

Comment 3: The authors should compare the LIP results in terms of imagery enhancements and processing time with the use of filters or arithmetic operations on channel values.

Answer 3: You are right. A novel section (6.3) has been dedicated to comparison with existing enhancement methods. Three figures (Fig. 8, 9 10) have been added. The execution time of the Full Dynamic Range Expansion algorithm has been estimated. Without specific optimization, an image with a standard resolution of 512x512 pixels is processed in approximately 0.032 seconds using a standard PC. (cf lines 259-260).

Comment 4; The 3D visualization of scanner dental optical cuts is a valuable application of LIP. However, the procedure for creating this visualization requires a clearer explanation.

Answer 4: This has been done in Section 7.1 “3D Visualization of Scanner Dental Optical Cuts” where Fig. 11 has been added and explanations given concerning the 3D block visualization.

Comment 5: The article outlines two main objectives, which should be explicitly stated as achieved in the conclusion.

Answer 5: This remark has been taken into account.

 Comment 6: The authors must divide the conclusion and perspectives section into discussion and conclusion sections.

Answer 6: I have divided the initial conclusion in two Sections :

8-Conclusion, in which the initial objectives are explicitely stated as achieved, according to your suggestion

9-Perspectives. In fact, the planned future work on the objective estimation of the visual quality of an image has already been undertaken and promising results have been produced.  

Submission Date

24 April 2024

Date of this review

27 May 2024 15:44:07